Low-level nutrient enrichment during thermal stress delays bleaching and ameliorates calcification in three Hawaiian reef coral species

Han Ji Hoon J. han9@hawaii.edu
Stefanak Matthew P.
Rodgers Ku‘ulei S.
Hawaii Institute of Marine Biology, University of Hawaii at Manoa , Kaneohe, Hawaii , United States
Banaszak Anastazia
Electronic publication date: 2022 Jul 14
Publication date: 2022
Volume: 10
Electronic Location ID: e13707
Received 2022 Jan 31; Accepted 2022 Jun 19
Copyright: © 2022 Han et al.
Copyright year: 2022
Copyright holder: Han et al.
License: This is an open access article distributed under the terms of the Creative Commons Attribution License, which permits unrestricted use, distribution, reproduction and adaptation in any medium and for any purpose provided that it is properly attributed. For attribution, the original author(s), title, publication source (PeerJ) and either DOI or URL of the article must be cited.
License URL: https://creativecommons.org/licenses/by/4.0/

Keywords: Thermal tolerance, Anthropogenic nutrient input, Bleaching, Coral reefs, Climate change

Funding: United States Geological Survey, Coastal and the Marine Geology Program’s cooperative agreement G13AC00130 This research was funded by the United States Geological Survey, Coastal and the Marine Geology Program’s cooperative agreement G13AC00130. There was no additional external funding received for this study. The funders had no role in study design, data collection and analysis, decision to publish, or preparation of the manuscript.

==============================
Terrestrial-based nutrient pollution has emerged as one of the most detrimental factors to coral health in many reef habitats. Recent studies have shown that excessive dissolved inorganic nutrients can reduce coral thermal tolerance thresholds and even exacerbate bleaching during thermal stress, yet the effects of minor nutrient enrichment under heat stress have not been extensively studied. In this study, Lobactis scutaria, Montipora capitata, and Pocillopora acuta colonies under heated conditions (~30.5 °C) were exposed to low and balanced nitrogen and phosphorous concentrations over a 31-day heating period. Coral colonies were collected from Kāne‘ohe Bay, O‘ahu, which has a unique history of nutrient pollution, and held in mesocosms that allowed for environmental manipulation yet are also influenced by local field conditions. Principal findings included delays in the bleaching of nutrient-enriched heated colonies as compared to heated-only colonies, in addition to relatively greater calcification rates and lower proportions of early-stage paling. Species-specific outcomes were prevalent, with L. scutaria demonstrating no difference in calcification with enrichment under heat stress. By the end of the heating stage, however, many heated colonies were at least partially impacted by bleaching or mortality. Despite this, our findings suggest that low levels of balanced nutrient enrichment may serve as a mitigative force during thermal events. Further field-based studies will be required to assess these results in different reef habitats.

Introduction

Coral reef systems are threatened globally by the pernicious effects of sea surface temperature (SST) rise, which can expose the coral holobiont to considerable thermal stress and potentially lead to mass bleaching events (Hughes et al., 2017). Coral bleaching, the process of major expulsion or digestion of zooxanthellae from or by the host, is instigated by the breakdown of the mutualistic relationship between coral and endosymbiont. These symbionts, along with host pigments in some species, give coral their characteristic coloration. The diminishment of color during bleaching thus results from the loss of these symbionts, which often then reveals the white skeleton beneath the now transparent tissues. Although the primary cause of widespread bleaching is considered to be heat stress (Hoegh-Guldberg et al., 2007), other large-scale chronic and acute factors exist that can induce and exacerbate reef-wide bleaching, including ocean acidification (Andersson et al., 2009; Jokiel, Bahr & Rodgers, 2013) and shifts in salinity, sedimentation, or irradiance (Easterling et al., 2000; Rodgers et al., 2021). Although repopulation by symbionts is possible under brief temperature deviations, sustained bleaching can lead to whole colony mortality (Jokiel & Coles, 1990). Given many coral species live within only 1–2 °C of their upper thermal limits (Coles, Jokiel & Lewis, 1976; Jokiel & Coles, 1977), observed increases in mean decadal SSTs are particularly harrowing given predictions of continual elevations in temperature due to climate change. If global mean surface temperature increases reach 1.5–2.0 °C above pre-industrial levels, warm-water reef mortality is expected to reach 70–90% worldwide (Hoegh-Guldberg et al., 2019).

The physiological mechanisms that drive coral bleaching have largely focused on photo-oxidative stress via photoinhibition of endosymbionts (Iglesias-Prieto et al., 1992; Lesser, 2006; Weis, 2008). The buildup of excess heat energy is partially transferred to oxygen, increasing concentrations of reactive oxygen species (ROS), which can cause substantial zooxanthellate cellular damage (Smith, Suggett & Baker, 2005; Weis, 2008), followed by the release of ROS into host cells (Downs et al., 2002) and eventual apoptosis (Dunn et al., 2004). More recent studies, however, have posited that amplified ROS-associated toxicity is at least preceded by independent host-specific immune responses to thermal stress (Krueger et al., 2015). Rädecker et al. (2021) found that temperature-associated increases in host energy demand may spur shifts in metabolism to reliance on amino acid catabolism, resulting in ammonium release and subsequent uptake by endosymbionts. This catalyzes algal growth, leading to decreased translocation of photosynthate to host cells and, ultimately, the degradation of symbiosis. These findings are particularly notable given that host mortality is thought to be driven at least in part by coral starvation (Anthony, Connolly & Hoegh-Guldberg, 2007).

Many local scale environmental factors also exist that can induce coral bleaching and mortality. Excess terrestrial nutrient input, which is commonly introduced through anthropogenic point sources (e.g., drain and sewage pipes) and non-point sources (e.g., impervious surface runoff, landslides), has been increasing due to shoreline development, agricultural activity, and storm frequency and intensity (Carlson, Foo & Asner, 2019; Giambelluca et al., 2013). Dissolved inorganic nutrients, such as phosphate (PO4−3), nitrate + nitrite (NO2− + NO3−), and ammonium (NH4+), are essential components in marine ecosystem functioning that govern biological productivity in marine environments (Smith, 1984). Most warm-water corals have evolved in nutrient-limited oligotrophic habitats, thus even minor augmentations of nutrient concentrations can lead to a variety of direct and indirect effects on coral health and metabolism.

Coral responses to nutrient elevation are differential, varying by species, local nutrient history, and nutrient type, concentration, and ratio (Bongiorni et al., 2003; Burkepile et al., 2020; Fabricius, 2005; Fox et al., 2021; Tanaka et al., 2017). A meta-analysis of historical nutrient research described how nitrogen enrichment oftentimes reduced calcification and enhanced photosynthesis in corals while, in contrast, phosphorous enrichment increased calcification with minimal effects on coral photosynthesis (Shantz & Burkepile, 2014). The role of nutrients in complex external and internal processes of coral holobiont metabolism, however, is not well understood.

Ammonium is the preferred source of dissolved inorganic nitrogen (DIN) for corals over nitrate and nitrite (Grover et al., 2003; Grover et al., 2008), which require reduction to be available for myriad metabolic processes. Several seminal laboratory studies have shown that exposure to relatively moderate concentrations of DIN (>2 µM) is associated with increased zooxanthellae density, chlorophyll concentration, and/or gross photosynthesis (Ezzat et al., 2019; Ferrier-Pages et al., 2000; Marubini & Davies, 1996; Muscatine et al., 1998; Stambler et al., 1991). These responses, however, are not necessarily indicative of healthy symbioses (i.e., reduced carbon translocation from higher endosymbiont population, see Dubinsky & Jokiel (1994)) and are inconsistent across different studies (perhaps due to variable experimental design), while oftentimes also accompanied by evidence of lowered calcification rates at higher concentrations (5–20 µM) of DIN (Fabricius, 2005). Moreover, it has been shown that dissolved inorganic phosphorous (DIP) exposure, although leading to higher growth rates, reduced skeletal density in new growth and thus affected structural quality (Dunn, Sammarco & LaFleur, 2012). More recent studies have found that, in the absence of concurrent DIP increases, sustained elevation of DIN concentrations may lead to phosphorous starvation and subsequent reduction in photosynthetic efficiency and carbon metabolism (D’Angelo & Wiedenmann, 2014; Ezzat et al., 2015; Rosset et al., 2017). Studies such as these have led to the development of a nutrient-response paradigm that is principally dependent on the nitrogen to phosphorous (N:P) ratio, which, if insufficiently balanced from DIN over-enrichment, can lead to degradation of the host-endosymbiont mutualism (D’Angelo & Wiedenmann, 2014; Morris et al., 2019; Tanaka et al., 2017). These studies, however, only applied nitrate as a DIN enrichment source, which is typically not as preferential as ammonium and generally causes deleterious effects at higher concentrations as compared to ammonium (Ezzat et al., 2015; Marangoni et al., 2020). On a community-wide scale, elevated nutrient input can stimulate excess macroalgal growth (Fabricius, 2005), which oftentimes leads to coral mortality via algal stressors such as overgrowth and shading (McCook, Jompa & Diaz-Pulido, 2001), inhibition of larval viability and settlement (Kuffner et al., 2006), and (or) allelochemical toxicity (Vieira et al., 2016). In cases of large-scale proliferation, lasting or permanent phase shifts from coral-dominant to algal-dominant regimes can occur, leading to shifts in community species composition and reductions in biodiversity (Adam et al., 2021; Duprey, Yasuhara & Baker, 2016).

The combination of elevated anthropogenically-sourced DIN and increased sea surface temperatures has long been considered as negatively synergistic with respect to coral health. Wooldridge & Done (2009) described how excess DIN exposure during thermal stress may preclude zooxanthellae from homeostatic growth-limitation, thus resulting in withholding of endosymbiont carbon translocation to the host during heat stress when it is particularly crucial, thus decreasing bleaching resiliency. Indeed, higher densities of algal symbionts have been shown to escalate susceptibility to bleaching (Cunning & Baker, 2013). Empirically, several field studies have linked elevated heat and anthropogenic nitrate to increases in bleaching susceptibility and intensity (Burkepile et al., 2020; Donovan et al., 2020). Furthermore, when considering ratio-dependent nutrient load, Wiedenmann et al. (2013) found that imbalanced increases in nitrate and concomitant phosphate starvation led to both greater susceptibility of bleaching during thermal stress and lowered thermal limits. Similarly, Chumun et al. (2013) observed more damage to endosymbionts under imbalanced high heat and nitrate conditions, in addition to impediment of recovery post-stress. Nitrate elevation-induced phosphate starvation may be especially harmful during heating given the finding that phosphate uptake rate is significantly higher during heat stress, which suggests an important role of DIP in bleaching resiliency during heating (Ezzat et al., 2016). While several studies treated nitrate (NO3−) as the main DIN source, recent studies have found that differential nitrogen identities (i.e., nitrate, ammonium, and (or) urea) can result in various physiological responses (e.g., Burkepile et al., 2020; Marangoni et al., 2020). Contrasting responses of bleaching intensity under thermal stress have indeed been found, with positive effects under ammonium (NH4+) enrichment versus negative effects of nitrate (NO3−) enrichments (Ezzat et al., 2019; Marangoni et al., 2020).

Despite many cases of decline, there have been some instances suggesting nutrients may have the ability to lessen or delay the effects of thermal stress under balanced ratio conditions (Wiedenmann et al., 2013). To our knowledge, this was first observed by McClanahan et al. (2003), where fertilizer containing high amounts of phosphoric acid, ammonium, and nitrate reduced paling from seasonal bleaching in treated colonies. Since then, Beraud et al. (2013) have found that balanced enrichment with ~3–3.8 µM ammonium during thermal stress resulted in an increased rate of photosynthetic activity and calcification when compared to heat-only treatments. More recently, when comparing the effects of nitrate and ammonium during heating, Marangoni et al. (2020) also recorded beneficial responses to 3 µM of ammonium with a balanced N:P ratio (~17:1) in the form of photosynthetic activity, calcification, and gauges of oxidative damage, although nitrate did not share these outcomes. Lastly, after minor yet chronic (~15 months) artificial increases in nitrate + nitrite, ammonium, and phosphate (on average 0.14, 0.14, and 0.22 µM, respectively), Becker et al. (2021) found improved holobiont performance under thermal stress, namely higher rates of maximal photosynthetic performance and oxygen evolution in nutrient-enriched groups. Despite several articles showing the potential of low nutrient enrichment to ameliorate various impacts of thermal stress, the type of enriched nutrients and N:P ratios largely vary by study and thermal stress period and were all restricted to less than 1 week. Although low enrichment under thermal stress has been shown to cause temporary changes in coral physiology that are beneficial, under longer periods of thermal stress these changes may be detrimental. There have been no published studies that have investigated the long-term effects of low nutrient enrichment under extended chronic thermal stress until this study.

To represent the prolonged duration of summer bleaching events in Hawai‘i more closely, we exposed three species of Hawaiian reef corals (Lobactis scutaria, Montipora capitata, and Pocillopora acuta) to increases of <1 µM of nitrate + nitrite, ammonium, and phosphate above ambient concentrations for 31 days under high temperature conditions. Of the species evaluated in this study, Pocillopora acuta has the lowest temperature threshold, followed by Montipora capitata (Jokiel & Brown, 2004; Ritson-Williams & Gates, 2020), while the more cryptic Lobactis scutaria is more resilient to elevated temperatures (Bahr, Jokiel & Rodgers, 2016); species that exhibit higher resistance to bleaching can be characterized by massive growth forms, thicker tissues, and heterotrophic feeding as in the solitary coral L. scutaria. Given these species-specific tolerances to heat stress, we expected differential responses to nutrient and (or) thermal stress. This study was undertaken using mesocosm tanks that create conditions comparable to the field yet allow for manipulation of seawater input (Jokiel, Bahr & Rodgers, 2014). Colony samples were collected from Kāne‘ohe Bay, which has a history of nutrient pollution, although substantial reductions in anthropogenic inputs have occurred following the relocation of sewage effluent offshore in 1979 (Smith et al., 1981). The high nutrient, anoxic, and low light conditions prior to sewage removal led to an extreme decrease in coral cover and a simultaneous increase in filter or deposit feeders, phytoplankton, and algae (Hunter & Evans, 1995; Laws & Redalje, 1982). This historical exposure to nutrient stress may have increased resilience to nutrient elevation in resident corals. Recent research has also shown that corals in Kāne‘ohe Bay are more acclimatized to thermal stress (Coles et al., 2018; Jury & Toonen, 2019) and may likewise be more resilient to nutrient stress. Response variables measured by visual assessment included partial and full colony bleaching and mortality. Calcification rates were quantified using the buoyant weighing technique and compared before and after heat exposure. Given outcomes from previously described studies, we expected that nutrient-enriched groups under thermal stress would show smaller decreases in calcification rate and reduced prevalence of bleaching and mortality during or at the end of the heating period.

Materials and Methods

Experimental design

The study period in this experiment spanned from June 21st to August 12th of 2019. A total of 360 coral colonies were collected, comprised of 120 individuals from three local species: Lobactis scutaria (Ls), Montipora capitata (Mc), and Pocillopora acuta (Pa).

All coral collections were authorized under the Division of Aquatic Resources Special Activities Permit SAP 2019-16. The reference to Pocillopora colonies as P. acuta, as opposed to their ostensible historical identification as P. damicornis, was implemented based on recent findings that P. acuta are commonly misidentified as P. damicornis in our study site of Kāne‘ohe Bay (Johnston, Forsman & Toonen, 2018). Colonies from each species were separated into four distinctive thermal and nutrient groups (Fig. 1), hereafter referred to by their abbreviations: a control group of ambient heat and nutrient levels (“A”), an ambient heat and elevated nutrient group (“N”), an elevated heat and ambient nutrient group (“H”), and an elevated heat and nutrient group (“NH”). As such, this led to a nested study design with mesocosm tank nested within treatment group and species. Target temperature increases in heat-treated groups were set at 31 °C. Ambient temperature and nutrient levels were representative of Kāne‘ohe Bay field conditions due to direct seawater influx.

Figure 1 Experimental design with sample sizes, and representative photograph of mesocosms.

Graphic nested study design (above) showing sample size of each treatment species and treatment group, with photograph of mesocosm system (below) depicting designated tanks by treatment (A = ambient heat and nutrients, N = ambient heat and enriched nutrients, NH = elevated heat and enriched nutrients, and H = elevated heat and ambient nutrient).

The study period was separated into consecutive acclimation and heating phases (Fig. 2). The initial acclimation phase lasted for 22 days (21 June–12 July 2019) with the intention of acclimating corals to mesocosms and, for nutrient-treated colonies, elevated nutrient concentrations. Following the acclimation phase, temperature was raised in heat-treated groups for 31 days (13 July–12 August 2019), which was defined as the heating phase. Nutrient elevation in treated groups was continuous throughout all phases. Measured temperatures and nutrient concentrations during each phase are shown by treatment group in Tables 1 and 2, respectively.

Figure 2 Timeline of study period.

Timeline of study period denoting phase periods and dates of data collection.

Table 1 Mesocosm water temperature by treatment group and experimental phase.

Water temperature (°C) measured from mesocosm water every 30 min. Mean ± SE temperature is shown by treatment group and experimental phase in addition to percentage of time tank temperature was ≥29 °C and ≥30 °C. Treatment group abbreviations: A (ambient heat and nutrients), N (ambient heat and enriched nutrients), NH (elevated heat and enriched nutrients), and H (elevated heat and ambient nutrients).

Temperature (°C)	A	N	NH	H	
Acclimation	28.03 ± 0.35	28.02 ± 0.39	28.03 ± 0.39	28.01 ± 0.35	
≥29 °C	10.2%	9.7%	9.5%	9.5%	
≥30 °C	0.3%	0.4%	0.2%	0.3%	
Heating	28.52 ± 0.51	28.48 ± 0.46	30.70 ± 0.52	30.73 ± 0.51	
≥29 °C	26.0%	25.6%	99.8%	100.0%	
≥30 °C	4.0%	3.8%	83.4%	87.7%	

Table 2 Mesocosm nutrient concentrations by treatment group and phase.

Nutrient concentrations (µM) measured from tank water every ~1 week. Above: comparisons of mean nutrient concentrations between aggregated enriched and ambient tanks (mean ± SE). Below: mean ± SE temperature by treatment group and experimental phase. Pairwise statistical comparisons of nutrient concentrations between treatment groups by phase are shown in Table S1.

Nutrients (μM) by treatment	Enriched	Ambient	Difference	p-value	
PO4−3	1.05 ± 0.77	0.39 ± 0.35	0.62	≤0.001	
NO−2 + NO−3	0.75 ± 0.45	0.09 ± 0.05	0.67	≤0.001	
NH4+	1.16 ± 0.64	0.23 ± 0.19	0.93	≤0.001	
Nutrients (μM) by phase	
Acclimation	
	A	H	N	NH	
PO4−3	0.207 ± 0.044	0.346 ± 0.294	0.784 ± 0.500	0.759 ± 0.284	
NO−2 + NO−3	0.226 ± 0.044	0.173 ± 0.019	0.524 ± 0.150	0.402 ± 0.105	
NH4+	0.238 ± 0.084	0.099 ± 0.021	0.550 ± 0.192	0.341 ± 0.058	
Heating	
	A	H	N	NH	
PO4−3	0.218 ± 0.117	0.570 ± 0.433	0.938 ± 0.484	1.213 ± 1.069	
NO−2 + NO−3	0.086 ± 0.053	0.089 ± 0.039	0.777 ± 0.435	0.721 ± 0.498	
NH4+	0.202 ± 0.155	0.261 ± 0.219	1.181 ± 0.609	1.132 ± 0.713	

Sample site and colony collection

Kāne‘ohe Bay is located in northeast O‘ahu, Hawai‘i (21°24′49.799″ N and 157°47′ 39.146″ W). It is the largest sheltered body of water in the Main Hawaiian Islands (MHI), approximately 28 km2 in size with a maximum depth of 12 m. Several reef structures can be found within the embayment. A fringing reef runs along the shoreline with over 50 patch reefs located within the lagoonal waters. The outer reef is composed mainly of limestone with a basalt foundation and is not a true barrier reef but is referred to as one.

Colonies of similar size and morphology of the three study species were collected from the Moku o Lo‘e shallow reef flat located adjacent to the research site at the Coral Reef Ecology Laboratory on the eastern point of the island (Fig. 3). All samples were collected from a depth of approximately one-half meter with colony size approximately 15 cm and limited to 200 g in weight. Colonies with partial damage or that showed signs of the disease were not collected. Conspecifics were gathered several meters apart to avoid genetic redundancy by fragmentation. Selected colonies were placed on underwater trays for transport and remained submerged during the transfer to laboratory tanks, where they were held for 1 week in gradually reduced shaded conditions to remediate any transfer-related stress. Following this respite period, 10 randomly selected colonies of each species were assigned to each of the twelve 660-L (1 m × 1 m × 0.5 m) fiberglass flow-through mesocosms for a total of 30 colonies per tank (Fig. 1).

Figure 3 Aerial image of sampling location and collection area.

Aerial image of colony collection sites near Moku o Lo‘e (red) and field nutrient sampling sites (yellow). Field site (South Bay and Intake Pipe) nutrient concentrations were compared with ambient mesocosm concentrations to assess uniformity between field and mesocosm conditions. Service Layer Credits: Esri, DigitalGlobe, GeoEye, Earthstar Geographics, CNES/Airbus DS, USDA, USGS, AeroGRID, IGN, and the GIS User Community.

Mesocosm flow-through system

The mesocosm tank system in this study modeled that of tanks originally designed by Jokiel, Maragos & Franzisket (1978) in order to closely replicate the environmental conditions (e.g., depth, light level, irradiance, etc.) of the reef flat collection site in Kāne‘ohe Bay. To limit fouling, water inflow was alternated through two intake pipes monthly. Additionally, the two separate lines transporting water from the intake pipe to the mesocosm system were reamed and switched monthly. Mesocosm water turnover (flow) rate was determined by closing the tail box drain and recording the time for water outflow to reach 10 L. Turnover rate was measured at least twice per day to ensure that all tanks had rates lower than 55 s/10 L (<60 min per complete turnover) to meet previously determined flow rate requirements described by Jokiel et al. (2008). To maintain water circulation, two Marineland® Maxi-Jet® 1200 multi-use water pump and powerheads were placed diagonally in the top corners of all tanks and set at the max rate of 295 gallons/hour. Tank oxygenation was sustained through an aeration stone that was centered at the bottom of the mesocosm. Water quality factors such as salinity, dissolved oxygen, and pH were monitored daily using a YSI-556 Handheld Multiparameter Water Quality Meter, although no anomalous conditions were detected over the course of the study period.

Abiotic factor manipulation and measurement

Finnex® TH Deluxe Titanium 800-Watt heating units were placed in all heat-treated mesocosms (NH and H) and were continuously active. Additionally, BlueLine Biotherm Titanium Aquarium 1,000-W heaters (Model IPX8) were also employed to reach target temperatures; all 1,000-W heaters were connected to Aqua Logic, Inc. Single Stage digital temperature controllers (model TR115SN), which activated 1,000-W heaters when temperature fell to 30.8 °C and deactivated heaters when temperature reached 31 °C. Heaters were placed in identical locations and orientations within each heat-treated mesocosm. Temperature was also recorded in all mesocosms during the entirety of the study period using calibrated Onset ProV2 HOBO automatic temperature loggers, which collected temperature estimates at 30-min intervals. Mean temperatures in the heated treatments (30.72 °C) (H, NH) were maintained at a level of ~2.2 °C above non-heated treatment groups during the heating phase (Table 1).

Nutrient concentrations in modified treatment groups (N and NH) were elevated by dissolving 39.879 grams of sodium nitrate (NaNO3), 33.465 grams of ammonium chloride (NH4Cl), and 29.799 grams of monopotassium phosphate (KH2PO4) into 100 L of ambient seawater from the adjacent reef flat. This resulted in an N:P ratio enrichment of approximately 1.94:1. Altered seawater was stored in a 100-L Nalgene™ cylindrical polypropylene reservoir tank and replenished using the same seawater to nutrient concentration ratios. To transfer nutrient-elevated seawater to selected mesocosms, a Cole-Parmer Masterflex® L/S Digital Drive EW-7523-80 was attached to two Cole-Parmer Masterflex® L/S Multichannel (model 7535-04) peristaltic pumps set at 5 mL/min. The peristaltic pump moved seawater through six 16″ Masterflex® Pump Tubings (model 96400-16), which were then connected to six less-flexible black polyethylene drip irrigation tubes (width 1.5 cm) that extended to the mesocosms.

Nutrient concentrations were measured weekly in each tank for the duration of the study period. Prior to sampling, twelve 1-L opaque Nalgene™ bottles were acid-washed using an 8% acetic acid solution. Bottles were then thrice rinsed with demineralized water and mesocosm water. To rinse, one-third of the bottle was filled then gently rotated and agitated for at least 15 s. Following rinsing, 1-L of mesocosm water was collected from same depth as coral colonies. A total of 100 mL of sampled water was filtered through a GF/F 25 mm Whatman glass microfiber filter into a 125 mL Nalgene™ bottle using a 60 mL syringe. A new filter was used for every water sample. Additionally, from the onset of the heating phase, bimonthly water samples were gathered at approximately 1 m in depth from field sites adjacent to the mesocosm intake pipe and the South Bay location (Fig. 3). Laboratory and field water samples were kept at −20 °C until their transport to the University of Hawai‘i School of Ocean and Earth Science and Technology Laboratory (S-LAB) for analysis. All nutrient concentrations were estimated using a Seal Analytical AA3 HR Nutrient Autoanalyzer. A detailed description of the instrument and specific procedures of the analyses for each nutrient compound can be accessed at http://www.soest.hawaii.edu/S-LAB/equipment/slab_autoanalyzer.htm.

Biotic factor measurement

Colony visual assessments were conducted every 2–3 days during the acclimation and heating phase. Visual assessment methodology was modeled from Jokiel & Coles (1974), which has since been used in modern studies (e.g., Coles et al. (2018)) and utilizes four visual categories to describe the partial condition(s) of individual colonies: normal, pale, bleached, and dead (Fig. 4). Colonies were carefully handled underwater to examine the colony’s entire surface, from which partial percentages of each condition were recorded (e.g., 0% normal, 40% pale, 40% bleached, 20% dead). If distinction between bleached and dead condition was unclear, a magnifying glass was used to determine polyp presence or absence. Other notable changes in condition were also documented, such as the percent of surface area tissue loss by disease, predation (Protestomia flatworms), or other parasitic organisms (e.g., algae, sponge, crab). 8% of M. capitata colonies were removed due to severe predator infestation that resulted in >20% tissue loss. Colonies with afflictions such as these were not included in any results or analyses.

Figure 4 Photographs of coral conditions for each study species.

Representative photographs of coral colony conditions used in visual assessments. Left to right: normal, pale, bleached, and dead. Top to bottom: L. scutaria, M. capitata, and P. acuta.

The calcification rate of each coral colony was evaluated at the end of the acclimation and heating phases. Calcification rate was measured using the buoyant weight technique described by Jokiel, Maragos & Franzisket (1978). Organisms that could affect colony weight (e.g., sponges, crabs, algae) were recorded then gently removed with seawater using a plastic syringe before measurement. Any broken coral fragments were also recorded and weighed with the colony. The measured colony weights were then converted to dry skeletal weights following procedures developed in Jokiel, Maragos & Franzisket (1978). Final calcification rate (mm/day) for a given phase was calculated as the cube-root of the difference in dry weights from initial to final measurements. From this method, calcification rate was expressed as a change in colony radial length as opposed to a simple change in weight, which allows for assessment of colony calcification differences irregardless of corallum size and morphology (Maragos, 1978).

Statistical approach

R (Version 3.6.2) and the integrated development environment, R-Studio Desktop (Version 1.1.453, RStudio PBC, Boston, MA, USA), were used for all data analyses (R Core Team, 2013). Significance level was set at α = 0.05 for all analyses. Graphs depicting coral survivorship, partial mortality, and partial bleaching were created using Microsoft Excel. All other graphs were created using R. Tank 8 (NH) was excluded from all abiotic and biotic data analysis due to malfunctioning of its temperature control systems. Furthermore, a total of 10 colonies with >30% predation of surface area were also removed from data analysis: A (6), N (1), NH (1), and H (2). A summary of sample sizes used in each biotic analysis are shown by phase, species, and treatment in Table S1.

Temperature data from 30-min intervals during the heating phase were compared between treatment groups to assess the effectiveness of artificial heating. Data were natural-log transformed to meet the assumption of normality and differences by treatment assessed using a one-way Analysis of Variance (ANOVA). Post hoc comparisons between treatment groups were subsequently evaluated by comparing Bonferroni-adjusted model estimated marginal means (EMMs) using the R package emmeans (Lenth et al., 2020).

The mean concentrations of nitrate + nitrite, phosphate, and ammonium from field sites (Intake Pipe and South Bay) were compared to those of the ambient nutrient group (A) over the entire study period (acclimation + heating phase). Homogeneity of variance for each nutrient was determined using Levene’s tests via the R package car (Fox & Weisberg, 2019). One-way standard or Welch’s ANOVAs, depending on the presence of homoskedasticity, were then used to compare differences in means between field sites and ambient mesocosm concentrations. Equivalent techniques were applied for pairwise comparisons between all mesocosm treatment groups during each phase. The mean concentrations of every nutrient type were also compared between aggregated ambient (A + H) and nutrient enriched (N + NH) treatment groups during the heating phase to confirm nutrient elevation during the experimental period. Unpaired two-tailed t-tests were used to compare these differences in means, with homoscedasticity assessed prior via Levene’s tests.

Flow rate data were averaged by tank over the entire study period and were found to be heteroskedastic. As such, analysis of flow rate differences between tanks was performed using a Welch’s one-way ANOVA, which does not assume equal variance between study groups (Welch, 1951). A Games-Howell (GH) post hoc test was utilized to compare individual contrasts in flow rate between tanks using the R package rstatix (Kassambara, 2021). Peristaltic pump rate over the entirety of the study period was also compared between tanks using a Welch’s one-way ANOVA, although post hoc comparison was not required.

Colonies were classified as dead when partial mortality was ≥95%. Binary survivorship data (1 = alive, 0 = dead) from the end of the heating phase and recovery phases were then compared using binomial generalized linear models (GLMs) with cloglog link functions and bias-reduced adjustments. The bias-reduced adjustment method is useful for binary data that show total or near complete uniformity in one or more factor levels, which occurred for some treatment groups with 100% or 0% survivorship and is available through the R package brglm2 (Kosmidis, 2017). Model factors used in binomial GLMs were coral species and treatment group, with post hoc contrasts between treatment groups compared among species using Bonferroni-adjusted EMMs. Model viability as compared to null models was assessed using Likelihood Ratio tests from the R package lmtest (Hothorn et al., 2020).

Colony partial bleaching and partial mortality proportion data at the end of the heating phase were modeled by beta regressions with logit link functions using coral species and treatment group as model factors. Proportion data were transformed using Eq. (1) (Smithson & Verkuilen, 2006) in order to account for zero and one data points, which cannot be incorporated into beta regressions. Equation variables n and y represent sample size and an untransformed proportion value, respectively.

(1) t(y)=y∗(n−1)+0.5n

Model viability as compared to null models were assessed using Likelihood Ratio tests and post hoc contrasts between treatment groups evaluated among species using Bonferroni-adjusted EMMs.

Days to bleaching (DTB) was determined as the number of days from the beginning of the heating phase to when colony partial bleaching reached ≥95%. No colonies from A and N treatment groups surpassed this threshold during the heating phase, thus DTB was only compared between NH and H. Unpaired Wilcoxon Rank-Sum tests were used to compare ranked DTB data between treatments among each species. Days to mortality (DTM), which was defined equivalently but instead in terms of partial mortality (≥95% cutoff), was also assessed using these statistical techniques.

Colony calcification data for all coral species were evaluated as the difference in mean calcification rate from the end of acclimation phase to the end of heating phase (referred to as “calcification-d”). Colonies with ≥50% partial mortality were excluded from statistical calcification comparisons due to the potential of dissolution. Mean calcification-d was compared between treatment groups using a linear mixed effect model with species as a fixed effect and treatment as both fixed and random. Viability of models were confirmed through assessment of residual plots. Post hoc differences between treatments were compared among species using FDR-adjusted EMMs.

Visual assessment data were assessed during Day 12 to examine corals near the experimental mid-point and reassessed at the end of the heating phase (Day 31). Individual colonies were assigned a single visual condition (i.e., normal, pale, bleached, or dead) determined by their most prominent visual condition. The proportions of specific visual assessment levels were assessed between pairs of treatment groups for each species. These comparisons were executed using a Chi-Squared Test of Proportions with Yates’ continuity correction, which is available through base R. To accompany quantitative and statistical analyses of visual assessment data, Multiple Correspondence Analyses (MCAs), which, similar to Principal Component Analyses, organize categorical variables in multidimensional factor space, were used to compare the explanatory contribution of different factor levels to spatial variation in data. MCAs were created and plotted using the R package FactoMineR (Lê, Josse & Husson, 2008).

Results

Environmental and abiotic parameters

Mean temperature varied by phase and treatment group (Two-way ANOVA, F(3, 51,213) = 13,208, p ≤ 0.001). Temperature in heat-treated groups exceeded 30 °C for >83% of the entire heating phase, as compared to only ~4% in ambient heat groups. After excluding heat-treated groups from the heating phase, temperature increased by an average of 0.48 °C from the acclimation phase to the heating phase. This nearly half degree C elevation in ambient water temperature over summer months coincided with seasonal changes in temperature associated with prolonged daylight and irradiance. It also occurred just prior to the 2019 bleaching event that peaked two months following the termination of this study. The island of O‘ahu experienced the lowest degree heating weeks (0.5 °C) during 2019 (Winston, 2022). Ambient temperatures did not exceed coral thermal thresholds during the experiment as evidenced by the lack of bleaching for corals under ambient conditions.

There were no observed differences in the mean concentrations of any nutrient type between field sites (South Bay and Intake Pipe) and the ambient mesocosms (A) over the entirety of the study period (Table S2), suggesting successful alignment of field and mesocosm nutrient conditions. When aggregating mesocosm nutrient treated (N + NH) and non-treated (A + H) groups, phosphate, nitrate + nitrite, and ammonium concentrations were on average 0.61 μM (t-test, T = 4.42, df = 48, p ≤ 0.001), 0.56 μM (t-test, T = 7.59, df = 36, p ≤ 0.001), and 0.75 μM (t-test, T = 6.74, df = 37, p ≤ 0.001) higher in enriched tanks, respectively, over the entirety of the study period. Comparisons of mean concentrations between all individual treatment groups are shown in Table S3.

Few differences in nutrient concentrations among nutrient enriched and ambient treatment groups were observed during specific phases. Mean ammonium concentration during the acclimation phase was greater in A than in H (t-test, T = 2.74, df = 7, p = 0.03) as well as in N than in NH (t-test, T = 2.49, df = 6, p = 0.05). During the heating phase, mean phosphate concentration was higher in H than in A (t-test, T = −2.79, df = 19, p = 0.01). Mean concentrations of all nutrient types by phase and treatment groups are shown in Table 2.

Mean flow rates (sec/10 L) were compared between individual tanks over the entire study period. The only significant differences in mean flow rates were between Tank 7 and Tanks 1 (Games-Howell, p = 0.01), 2 (Games-Howell, p = 0.004), and 12 (Games-Howell, p ≤ 0.001), although the mean increase in flow rate by Tank 7 only ranged from 8.4–9.8 s/10 L (Welch’s One-way ANOVA, F(10, 297) = 3.96, p ≤ 0.001). All tanks had mean flow rates lower than 60 s/10 L (<60 min per complete turnover) in accordance with methods stipulated by Jokiel et al. (2008) (Table S4).

The peristaltic pump rate (mL/min) of nutrient input into individual mesocosm tanks was also assessed over the entire study period. There were no differences in pump rate between any tanks, as mean rates ranged from only 3.17–3.24 mL/min (Table S5).

Survivorship and partial mortality

Coral survivorship and colony partial mortality did not differ by treatment group at the end of the heating phase. The lowest observed survivorships were P. acuta (84%) and L. scutaria (85%) in NH treatment groups (Table S6); mean partial mortality also reached as high as 22.4% and 21.3% in H and NH P. acuta colonies, respectively, although these changes were not significantly different from other conspecific treatments groups (Table S7).

Partial bleaching

Partial bleaching was assessed over the course of the heating phase as a more sensitive measure than total and partial mortality due to low mortality at the end of the heating phase (Fig. 5). Partial bleaching at the end of the heating phase varied by treatment group (Beta Regression Model, pseudo-R2 = 0.74, Z(13, 314) = 215.74, p ≤ 0.001) but not by species. For all species, partial bleaching was higher in NH and H treatments than in A and N treatments (EMMs, see Table S8). There were no differences in partial bleaching among unheated groups (A and N) and heated groups (NH and H), except for L. scutaria, which showed higher partial bleaching in H than in NH (EMM, p ≤ 0.001, Table S8).

Figure 5 Partial bleaching and mortality during the heating phase.

Mean ± SE partial bleaching and mortality (%) during the heating phase for NH (black lines) and H (red lines) treatment groups shown by species. Dotted lines depict partial bleaching whereas solid lines represent partial mortality.

Days to bleaching/mortality

DTB was not assessed in A and N treatment groups as no colonies reached the ≥95% partial bleaching threshold that was selected to constitute full colony bleaching. Among heat treated groups, however, DTB was lower in H than it was in NH for all species (Wilcox Test, p < 0.05 for all comparisons, Table S9). On average, bleaching began 3.4 days earlier in H treatment group colonies as compared to those of NH across all species.

No coral mortality occurred in the ambient nutrient group (N) and only one colony (M. capitata) died under ambient conditions (A), thus DTM was not evaluated for N and A treatment groups. No significant differences in DTM were found between NH and H treatment groups among any species (Table S10).

Calcification rate

The change in calcification rate (calcification-d) from the end of the acclimation phase to the end of the heating period was also assessed between treatment groups among species. Minor changes at the end of heating period were observed for ambient heating groups, although overall decreases in calcification rate occurred for both NH and H (Fig. 6). Calcification-d in A and N treatment groups was higher than calcification-d in H treatment groups among all species (EMMs, p < 0.05 for all comparisons, Table S11), with no differences detected between the A and N control groups. Calcification-d differences between both control groups (A and N) and NH were found in M. capitata colonies (EMMs, p < 0.05 for both comparisons, Table S11), as calcification-d in the NH treatment group was only marginally greater than that of the H treatment group. There were no significant differences between H and NH treatment groups for L. scutaria and M. capitata, although calcification-d was lower in H than in NH for P. acuta colonies (EMM, p ≤ 0.001, Table S11). These findings placed NH as an intermediate treatment group between ambient heat groups and H with respect to calcification-d despite the overall decrease in calcification rate at the end of the heating period in all species (Fig. 6).

Figure 6 Calcification rate by treatment group and species.

Mean change in calcification rate (mm/day, “calcification-d”) from the end of acclimation phase to end of heating phase by treatment and species. Pairwise comparisons of calcification-d between treatment groups are shown using brackets: NS = not significant, (*) = p ≤ 0.05, and (***) = p ≤ 0.001).

Visual assessments

By Day 12, the proportion of paled colonies in H treatment groups were greater than those in all other treatment groups across every species (Chi-Squared Tests, p < 0.05 for all comparisons, Table S12), except between NH and H treatment groups of P. acuta. Despite this, however, 83% of P. acuta H colonies were either paled, bleached, or dead, as compared to 53% in NH colonies. For all species, the percentage of normal colonies did not differ between A and NH treatment groups, whereas normal colony percentage was greater in N than in A and NH treatment groups excluding P. acuta (Chi-Squared Tests, p < 0.05 for all comparisons, Table S12). Percentages of visual assessments by treatment and species are shown in Fig. 7.

Figure 7 Visual assessment on day 12 of heating between treatment groups by species.

Percentage (%) of colonies that were either normal, pale, bleached, or dead by treatment group and species on Day 12 of the heating period.

The Multiple Correspondence Analysis of visual assessments from Day 12, which explained 35.3% of total variation, demonstrated several of the same quantitative patterns in Euclidean space. Treatment groups N and H were distinctly separated in first dimensional space (x-axis), whereas A and NH were located at an intermediate distance (Fig. 8, see Table S13 for exact coordinates). These spatial trends were driven by normal and pale colonies, which explained 21.7% and 21.0% of first-dimension variation, with the normal condition centered negatively and the pale condition centered positively. Coral species contributed minimally to variation and was centered similarly for each species.

Figure 8 Multiple Correspondence Analysis (MCA) plot by treatment group on day 12.

Multiple Correspondence Analysis (MCA) plot of treatment, species, and visual assessment data from Day 12 of the heating period. Treatment group centroids are shown along with corresponding 95% CI ellipses.

At the end of the heating period (Day 31), the bleached colony percentage of L. scutaria was lower in NH than in H (Chi-Squared Test, χ2 = 10.09, df = 1, p ≤ 0.001), whereas no differences in the M. capitata and P. acuta colonies of these treatment groups were observed (Table S14). The percentage of paled colonies of L. scutaria was also lower in the N than the A treatment groups (Chi-Squared Test, χ2 = 8.42, df = 1, p = 0.004), although this was again not observed for other species. Altogether, there were no recorded bleached or dead colonies in the A and N treatment groups (Fig. S1).

The MCA of visual assessments from the end of the heating period well represented data in two-dimensional space (41.3% cumulatively, Fig. S2) and visualized similar trends as were described above. The model’s first dimensional space showed separation of heated and non-heated treatment groups, which was driven predominantly by the normal and bleached conditions (see Table S15 for exact coordinates). Alternatively, the variation in second-dimensional space was primarily determined by the pale condition, with the A and NH treatment group centered more positively. This may be concomitant with the lower percentage of bleached corals in NH than in H of L. scutaria colonies (Fig. S1), which may instead only have paled, suggesting this trend may be driven only by L. scutaria. Indeed, there were differences in species position in second-dimensional space, with L. scutaria more positively centered in comparison with other species.

Discussion

In all examined coral species, enrichment of heat-stressed colonies with low levels of inorganic phosphorus and nitrogen concentrations resulted in improvements of coral conditions indicative of maintaining the host and endosymbiont relationship. In this study, we conducted mesocosm-based experiments on coral exposure to minimal balanced nutrient enrichment (<1 µM of nitrogen and phosphorous) during a prolonged heating event (31 days), which are becoming increasingly common in sub-tropical environments such as the Hawaiian Islands. We assessed colony growth and various coral conditions over the study period (i.e., survivorship, partial mortality and bleaching, bleaching progression), finding evidence of ameliorated growth rates and delayed bleaching in colonies subjected to thermal stress and nutrient enrichment. For all species tested, the onset of bleaching was delayed by 3.4 days on average in nutrient-enriched and heated colonies as compared to heated-only colonies, which was likely due to the reduction in paled individuals midway through the heating period (Fig. 7). Relatedly, bleaching was consistently lower proportionally in nutrient-enriched L. scutaria and M. capitata colonies during the heating period (Fig. 5), albeit a lack of statistical difference during the final day of heating. This study is the only published account of prolonged coral exposure (>30 days) to elevated temperature and minimal balanced nutrient enrichment, expanding on shorter duration studies (e.g., Ezzat et al., 2016; Hadjioannou et al., 2019; Tanaka et al., 2014) to investigate chronic exposure, which may more closely resemble thermal events in the field.

Calcification rate, which was used as a proxy for host bioenergetic and metabolic status, was also less affected in nutrient-enriched groups during heat stress. The enhancement of host metabolism under enriched nutrient conditions has been recorded in previous studies (Beraud et al., 2013; Fabricius, 2005; Tanaka, Hayashibara & Ogawa, 2007) and may help to explain relatively lower calcification rates of NH as compared to H treatment groups observed in this study. Altogether, these results suggest that minor and balanced nutrient enrichments can increase the resilience of individual colonies to thermal stress during heating events.

The implementation of Multiple Correspondence Analyses (MCA) to visualize patterns in categorical datasets has been utilized in ecological studies (e.g., Aubert et al., 2006; D’Onghia et al., 2011), although it remains somewhat irregular as oftentimes biological sampling predominantly generates numerical response data. Here, however, our visual assessments were modeled after techniques employed by Jokiel & Coles (1977) and subsequently Coles et al. (2018), which permit the rapid and practical evaluation of the bleached status of large numbers of coral colonies every 2–3 days, producing categorical data. The combination of descriptive MCA plots with subsequent chi-squared proportional tests allowed for visual and statistical interpretation of these data. This approach revealed that nutrient enrichment decreased bleaching of heated colonies during the intermediate stage (Day 12) of the heating period. By the end of heating (Day 31), the nutrient-enriched heated group had higher proportions of paled colonies than did the heated group, which were instead primarily bleached or dead. These indications of remediated health in nutrient-treated groups at the end of heating are mostly attributed to L. scutaria, however, as paled colony proportion did not vary significantly among other species.

The delayed decline in colony visual conditions may be tied to the qualitative observation of greater zooxanthellate density in nutrient-enriched colonies in both heated and unheated groups. Although over-proliferation of microalgal symbionts under high and unbalanced nutrient enrichment has been shown to exacerbate bleaching (D’Angelo & Wiedenmann, 2014; Rosset et al., 2017; Wooldridge & Done, 2009), the modest concentrations of balanced nutrient enrichment in this study may have been too low to cause overgrowth and thus minimized any associated negative effects. This aligns with expectations expressed by D’Angelo & Wiedenmann (2014) who proposed that N/P concentrations may only be harmful under imbalanced ratios. This interpretation is further supported by new evidence that endosymbionts can maintain stable host-symbiont nutrient relationships during modest fluctuations in density due to alterations in symbiont photophysiology (Krueger et al., 2020). Although we lack measurements of causal variables such as photosynthetic rate or cellular nutrient composition that could confirm these conditions, the nutrient concentrations in our mesocosm experiments were similar to measurements recorded by Becker et al. (2021) in their field manipulation study, which observed increases in gross photosynthetic rate and maximal performance. Our observations of mollified calcification rates may also be indicative of host metabolic mitigation as past studies have observed greater skeletal extension of corals when enriched with phosphate (Dunn, Sammarco & LaFleur, 2012; Shantz & Burkepile, 2014), although the structural integrity of new growth may be compromised. As such, the presence of phosphate in our nutrient treatments may have been the main driver of observed improvements to visual health and calcification rate due to sufficient supply of phosphate for prevention of P-limitation (D’Angelo & Wiedenmann, 2014), whereas the negative effects from nitrate and nitrite may have been minimized or otherwise influenced by the presence of ammonium (Ezzat et al., 2016; Marangoni et al., 2020). The duration of positive effects from phosphate introduction is unclear, however, as bleaching was quite pervasive by the end of the heating period.

Several past studies have summarized potential mechanisms for augmented coral resiliency in response to chronic or multiple acute disturbance events (Suggett, Warner & Leggat, 2017; Torda et al., 2017). This “ecological memory” (see Hackerott, Martell & Eirin-Lopez, 2021; Hughes et al., 2019) approach is a leading rationale supporting current and future individual and community-level resiliency in the context of anthropogenic stressors. Tolerance to higher temperatures and reduced water quality, for example, have been observed in reef habitats over generational scales (Coles et al., 2018; Tisthammer et al., 2021), with more recent studies suggesting that rapid thermal acclimatization may occur in response to acute environmental stresses through non-genetic inheritance, epigenetic adaptation, and (or) microbiome plasticity (Littman, Willis & Bourne, 2011; Palumbi et al., 2014; Putnam, 2021). Kāne‘ohe Bay, which was historically highly suitable for reef coral growth (Edmondson, 1928), entered a period of severe coral decline after introduction of major high-nutrient sewage output in 1963. After the majority of sewage flow was removed in 1978, Kāne‘ohe Bay underwent a reverse phase-shift to a coral dominant regime over the next several decades (Smith et al., 1981). Initial coral recovery was relatively immediate, with reported increases in coral cover of 26% within several years of effluent removal (Smith et al., 1981). It has also been suggested that corals in Kāne‘ohe Bay have developed improved thermal tolerance due to the incumbent high temperatures relative to adjacent reefs (Coles et al., 2018; Jury & Toonen, 2019) and, when considering nutrient history in tandem, may have the potential for improved tolerance to combined nutrient and heat augmentation. Accordingly, nutrient history has recently been shown to mediate the response of Scleractinian corals to thermal stress (Hadjioannou et al., 2019).

The differential responses to heat stress by the three coral species in this study with respect to visual assessment, calcification rate, and partial mortality and bleaching may be due to differences in life history, morphology, and stress tolerance. P. acuta, which declined rapidly in visual status and partial mortality, also demonstrated the greatest decrease and variation in calcification rate when exposed to heat stress and nutrient enrichment (Fig. 6). General trends suggest that L. scutaria may be more tolerant to thermal stress (Bahr, Jokiel & Rodgers, 2016; Coles et al., 2018) with and without nutrient enrichment, as these colonies exhibited less bleaching during the intermediate periods of the heating phase and showed no statistical differences in calcification rate between the NH treatment group and ambient groups. When considering variation in response, factors that could contribute to species-specific performance are heterotrophic feeding plasticity and endosymbiont regulation. Coral heterotrophic feeding has been recognized as one of the main sources for energy acquisition during stressed conditions, especially under disruption of endosymbiont autotrophy (i.e., bleaching) (Grottoli, Rodrigues & Palardy, 2006). Greater reliance on heterotrophic feeding can alleviate bleaching and maintain coral growth by fulfilling carbon demand that is lost from the breakdown of symbiosis (Houlbrèque & Ferrier-Pagès, 2009). A recent study by Price et al. (2021) suggested that the average contribution of heterotrophy to Hawaiian coral feeding is approximately 20–50%, depending on the species. The level of heterotrophic feeding was likely variable among the coral species examined in this study and may have confounded the magnitude of responses to heat stress. Indeed, our results showed delayed bleaching (Days to bleaching, DTB) was more pronounced in M. capitata (Fig. 5, Table S9), the species with the highest heterotrophic feeding rate during post-bleaching recovery. This concurs with other studies that report M. capitata was able to obtain up to 100% of the required metabolic energy requirements from heterotrophy (Dobson et al., 2021; Grottoli, Rodrigues & Palardy, 2006). Alternatively, the relatively poor response of P. acuta to heating in this study may have been driven by its limited ability to regulate endosymbiont proliferation (Fox et al., 2021). These results imply that any positive effects of balanced nutrient enrichment may be mediated by the stress tolerance of the affected species.

Our study presents evidence for possible bleaching mitigation under low-level nutrient enrichment. Temperature, however, was unequivocally the dominant factor that induced bleaching, considering there were clear differences in all measured parameters between ambient (~28.5 °C) and heated (~30.7 °C) groups regardless of their nutrient regime. Although not measured in this study, the Symbiodiniaceae genotype may have prompted differences in visual conditions and calcification, as several articles have shown thermal tolerance differences between Symbiodiniaceae genotypes (Grégoire et al., 2017; Stat & Gates, 2011). Moreover, the variations in nutrient uptake, carbon translocation rates, and nutrient competition between Symbiodiniaceae genotype have been found (Baker et al., 2013), and it is plausible that the Symbiodiniaceae species competition (dominance) could be altered under nutrient enrichment due to differing nutrient processing capacities. The effect of nutrient enrichment on Symbiodiniaceae competition and the impact on thermal tolerance requires further investigation.

Although the utilization of a mesocosm study system can control for many factors that cannot be regulated in traditional field studies, there remain some limitations. Adjustment of nutrient inputs into enriched tanks required consistent modification, as uptake and release of measured nutrient concentrations by corals continuously alters in situ tank concentrations. This led to higher ranges in measurements, as was reflected in the standard error of nutrient concentrations, although this is a common issue in nutrient manipulation studies (e.g., Fox et al., 2021) and did not preclude statistically significant differences between enriched and ambient treatment groups. Secondly, because mesocosms rely on proximate seawater influxes, the water temperature of ambient tanks during the heating period rose on average to sub-lethal levels (~28.5 °C). Interestingly, however, little paling or bleaching was observed in nutrient-enriched ambient-temperature colonies as compared to fully ambient colonies (Fig. S1), which is consistent with observations by McClanahan et al. (2003) that fertilizer enrichment precluded typical summer bleaching. As such, the mitigative potential of nutrient enrichment in this study may also apply to corals experiencing sub-lethal temperature exposure.

Conclusions

Despite clear negative feedbacks of eutrophication on reef degradation (Littler & Littler, 1984; Wooldridge, 2013), deciphering the effects of direct nutrient enrichment on coral thermal tolerance has been a convoluted and ambiguous initiative for several decades. Dissolved inorganic nutrients should be treated as a “double-edged sword” that can alter coral holobiont symbioses (Wooldridge, 2013). Many studies have found that excess nitrogen can exacerbate bleaching in field and laboratory settings (e.g., Burkepile et al., 2020; Chumun et al., 2013; Nordemar, Nyström & Dizon, 2003). In this context, both nutrient identity and N:P ratio have been identified as controlling factors influencing coral response (Ezzat et al., 2015; Marangoni et al., 2020; Rosset et al., 2017; Wiedenmann et al., 2013). Therefore, the conditions and threshold that balances beneficial and harmful impacts of nutrients on coral is crucial to elucidate for near-shore management and maintenance of healthy reef ecosystems in a changing ocean. Although critiques of past studies have posited that coral exposure to large N:P ratios and (or) concentrations does not occur in natural systems, recent studies have demonstrated that increases in N:P ratio at tested levels can occur via anthropogenically driven nutrient enrichment or excessive runoff (Lapointe et al., 2020), which has coincided with bleaching events (D’Angelo & Wiedenmann, 2014; Lapointe et al., 2019). While the importance of nutrient availability (N:P ratio) to coral thermal bleaching has been discussed, the coral host can limit or stabilize the algal proliferation (Baird et al., 2009) and thresholds of nutrient concentrations and (or) N:P ratios that balance the presence of mutualism and parasitism have not yet been ubiquitously identified. Here, we presented novel results that infer low and balanced nutrient enrichment during thermal stress may mitigate coral bleaching during the early stages of a heating event and help to maintain mutualism. Our findings seemingly align with those of Becker et al. (2021) and are particularly relevant considering many reef systems are experiencing sustained or increased anthropogenic nutrient input from sources such as fertilizer and sewage outfall, which can lead to macroalgal overproliferation and associated coral mortality. Determining the remediatory potential of this approach across different spatial scales will require similar mesocosm or field studies in other reef systems that consider local nutrient dynamics, as Kāne‘ohe Bay is unique in its nutrient and temperature history.

Supplemental Information

Supplemental Information 1 Supplemental figures and tables.

Click here for additional data file.

Supplemental Information 2 Full raw data.

Click here for additional data file.

This work builds upon the research foundation of Paul L. Jokiel, a pioneer in the field of Marine Science. Contributions to experimental maintenance and design were made by Dr. Evelyn Cox, Dr. Steve Coles, Andrew Graham, Anita Tsang, Sarah Severino and Yuko Stender of the Coral Reef Ecology Laboratory at the Hawai‘i Institute of Marine Biology. The authors would also like to thank Dr. Jake Ferguson for his guidance regarding several statistical techniques. This is HIMB contribution number: 1884 and SOEST contribution number: 11490.

Additional Information and Declarations

Competing Interests

Author Contributions

Field Study Permissions

Data Availability

The authors declare that they have no competing interests.

Ji Hoon J. Han conceived and designed the experiments, performed the experiments, analyzed the data, prepared figures and/or tables, authored or reviewed drafts of the article, and approved the final draft.

Matthew P. Stefanak analyzed the data, prepared figures and/or tables, authored or reviewed drafts of the article, and approved the final draft.

Ku‘ulei S. Rodgers conceived and designed the experiments, performed the experiments, authored or reviewed drafts of the article, and approved the final draft.

The following information was supplied relating to field study approvals (i.e., approving body and any reference numbers):

The Department of Aquatic Resources granted Special Activities Permit SAP 2019-16 to carry out this study.

The following information was supplied regarding data availability:

The raw data are available in the Supplemental File.

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
