# Peer review of "Low-level nutrient enrichment during thermal stress delays bleaching and ameliorates calcification in three Hawaiian reef coral species"

_PeerJ, doi:10.7717/peerj.13707_

## Round 0.1 · original submission · Major Revisions

I have received thorough evaluations from three reviewers and their comments can be seen below. Please address these comments in a revised version of the manuscript.

Reviewer 1 ·

Basic reporting

The authors used professional English through the manuscript, which was well-written with clear research goals and hypotheses. Ample references to relevant literature were included. The structure of the manuscript followed standard guidance for PeerJ research articles, and the raw data was provided. See general comments for suggested edits.

Experimental design

This experiment investigated the bleaching and mortality response of three coral taxa in relation to elevated heat stress and nutrient input in a controlled laboratory setting. The authors designed a study that address a knowledge gap in how coral responds to low and balanced nutrient enrichment during heat stress. Given the forecast of rising ocean temperatures and increasing severity and frequency of coral bleaching events over coming decades, this work is relevant and timely with the potential for informing local management. The methods of this manuscript are extremely detailed and contain enough information for this experiment to be replicated.

Validity of the findings

I believe the offers used appropriate statistical methods throughout their analysis, and supported their test choices with appropriate citations. The underlying data used in the analysis is provided – please see my notes under general comments for suggestions regarding the formatting and presentation of the data. The authors conclusions addressed their original research aims and sufficiently backed up relevant literature.

Additional comments

Please see comments below per section of the manuscript –

Introduction: The authors give a very thorough background on the physiological mechanisms of coral bleaching and the local environmental drivers of this process, with a strong focus on nutrient input. Given this study did investigate the response of three different taxa, I suggest that the introduction describe these three species and what is known of their varying bleaching thresholds.

Moreover, Kāne'ohe Bay is indeed a unique environment with a dynamic history of nutrient input. While the authors address this in the discussion section of the manuscript, I think more should be stated on this matter in the introduction. The introduction is quite limited in describing Kāne'ohe Bay’s nutrient history – lines 125-127. I suggest that the authors clearly state that the coral here have been posited to be more resilient to thermal and nutrient stress. I see this as an important lens to interpret this study through.

Methods: I appreciated the structure of this section. However, after reading the ‘Biotic Factor Measurement’ description, I did go back to lines 128-129 of the Introduction. If I am interpreting the methods correctly, observations of bleaching and mortality, and days until bleaching or mortality, were determined via visual assessments. Therefore, I recommend that the authors edit lines 128-129 for clarity. In lines 239-249, the authors state that notable changes in condition were documented; in reviewing the data, it was unclear to me where I could find this information.

Results: The authors report in the lines 341-343 that the rise in ambient temperature coincided with “expected seasonal changes in temperature” – however, sea surface temperatures in Hawaii were tracking well above average during the summer of 2019 and preceded a mass bleaching event during the autumn months.

Acknowledgements: The PeerJ guidance states: “Do not acknowledge funders here, there is a separate Funding Statement for that” – revise accordingly.

Tables:

Table 1 contains a wealth of information, which the caption does not quite accurately describe – the caption should be revised. Consider breaking into two tables (temperature and nutrients). The last section showing nutrients by phase is also slightly challenging to read as it lacks a clear distinction between acclimation and heating. The treatment groups (A, N, NH, and H) should also be described within the first table and first figure caption in the manuscript for ease of interpretation.

Table 3 may not be necessary – consider reporting this information in the text to cut length. This information is technically also visualized in Figure 4.

Figures:

Figure 2 – I would recommend improving image quality here and cite the source of the imagery in the figure caption.

Figure 4 is a good representation of the trajectory of bleaching for each of the taxa of interest – but is it showing both ‘paling’ and ‘bleaching’ or just ‘bleaching’ (line 234). Given bleaching remained at 0% for the Ambient and Nutrient only treatments, I would recommend removing these from the figure in favor of adding partial mortality over time per species.

Figure 5 – There seems to be room for the species names to be spelled out (as in Fig. 4); I suggest doing that here or describing the codes in the figure caption.
The use of MCAs is interesting, but ultimately I find figures 6 + 7, and 8 + 9, to be redundant. I suggest moving one set of the figures to the supplementary material. Moreover, figures 6 and 8 do contain data that is partially shown in Figure 4. If Figure 4 was revised to show changes in paling and/or partial mortality over time, or if this information is already included in a table, perhaps figures 6 and 8 could be removed from the manuscript.

Data: In the dataset key, why two different code options for the treatments? I recommend that this dataset come with a full metadata sheet, where units and descriptions are each set of data are provided. Metadata should also contain dates that observations were recorded.

The data includes information on ‘days to mortality’, but this is not reported in the manuscript.

I would consider renaming the ‘Visual Assessments’ sheet to better describer that data, since other data was also collected through visual observation.

What do ‘*’ represent in the data under ‘Number’?

Reviewer 2 ·

Basic reporting

This is a useful set of data and worth publishing. However, there is room for improvement and the comments included below should be addressed prior to publication. In general, the language / expressions of the ms should be subjected to an in-depth editing run. The ms is overillustrated and a number of figures / could be lost / moved to the supplement. There are concern regarding the interpretation of the “calcification rates” and some of the statements discussion items and citations need to be thoroughly revised.

Title “Low nutrient enrichment during thermal stress” –
Sounds a bit awkward. Use “low-level” or “mild instead”?


“demonstrating the greatest remediation from enrichment under heat stress.”
Unclear / awkward – replace or rephrase remediation in this context.


“Visually, this is generally 21 characterized by the paling and eventual loss of pigment, resulting in a transparent appearance and 22 revealing the skeleton below the tissues.”
Unclear, re-write / formulate in a way that reflects the different ways by which bleaching may occur.


“drive coral bleaching are still somewhat abstruse”
awkward, rephrase “abstruse”. Actually, quiet a lot is known. Describe the known mechanisms.


will lead to degradation of the host-endosymbiont mutualism (D’Angelo & 79 Wiedenmann 2014; Morris et al. 2019; Tanaka et al. 2017).
Cite Wiedenmann et al. (2013) Nature Climate Change in this context, which was the first paper to outline this concept.


Despite many cases of exacerbation, there have been some instances suggesting nutrients
104 may have the ability to placate the effects of thermal stress under balanced ratio conditions
Awkward / unclear: Rephrase “exacerbation” and “placate”


51 landslides), has been increasing due to forces such as shoreline develop
Rephrase “forces” in this context.

“106 amounts of phosphoric acid, ammonium, and nitrate showed an ancillary association”
Rephrase “ancillary”

“in mitigation109 of photosynthetic activity”
This sentence does not work, replace mitigation.

111 recorded beneficial responses to 3 μM of balanced ammonium
balanced ammonium? – unclear


Despite 117 several articles showing the potential of low nutrient enrichment to ameliorate various impacts of 118 thermal stress, there have been no published studies known to the authors that have investigated 119 the effects of balanced low-level enrichments of DIN or DIP on chronic thermal stress (> 2 weeks 120 of exposure).

The logic of this sentence is not clear. Are these results published or not? Make clear how the present study is different from past ones.


elevations of nitrate + nitrite, ammonium, and 123 phosphate
Rephrase “elevation”


As designed, mean temperature varied…
Rephrase “as designed”


“Temperature in the NH and H treatments during the heating phase, 337 which reached means of 30.70 °C and 30.73 °C, respectively, was higher from those of the non-338 heated treatment groups (Table 1).”

This is not really a result, but a confirmation that the setup works. Suggest to shorten this and integrate in the method section. Mention the conditions (also the control temperature!) in the results a brief introduction when talking about the effects of the treatment.


“There were no observed differences in the mean concentrations of any nutrient type
345 between field sites and the ambient mesocosms over the entirety of the study period (Table S2),”

Unclear, this is for the control condition? The enriched systems should show some elevated values? Table 1 seems to suggest this?
Table on should use terminology consistently, the concept “actual and 3 target enrichment.” Cannot easily follow through the table. Also, it is unclear what the difference between “nutrients” and “nutrients by phase” is.


• Some tables contain redundant information compared to the graphs. Try to move as many as possible to the supplement to reduce the overall (too high) number of display items.

• The MCA plot analysis does not contribute a lot information that goes beyond the more accessible graphs, and can probably be left out.

• Figure 6 and 8 should be combined in a meaningful way.


Figure 5: Remove residual dots for the axis label. Not sure if I read this correctly, but there seems to be no calcification, also under control conditions, but rather a weight loss in the stressed colonies, that is less pronounced in the nutrient enriched samples. If this is not the correct / intended interpretation, more explanations are needed. Accordingly, it seems wrong to talk about increased calcification rates. The relevant sections need to be rewritten throughout the ms.


“This study is the first published 445 mesocosm-based account known to the authors that has shown coral exposure to minimal balanced 446 nutrient enrichment (<1 μM of nitrogen and phosphorous) during”

Remove priority claims, in particular if you don’t know for sure if they hold.


“the modest concentrations of balanced nutrient enrichment 479 in this study may have been too low to cause overgrowth and thus minimized any associated 480 negative effects.”

According to D’Angelo & Wiedenmann 2014, detrimental effects would be only expected if the N:P ratio is not favourable (P-starvation). High N/P concenctrations were not necessarily harmful if available in a permitted ratio. You data confirm this nicely.


“with more recent studies even suggesting rapid adaptation is possible in 505 response to acute stress due to clade changes and/or (epi)genetic adaptation (Buerger et al. 2020; 506 Dixon et al. 2015; Palumbi et al. 2014).”

This needs to be toned down and discussed more critically, there is limited evidence that many species can easily change clade / adapt. See Goulet 2006 DOI:10.3354/meps321001


Many studies have found that excess 555 nitrate can exacerbate bleaching in field and laboratory settings (e.g., Burkepile et al. 2020; 556 Chumun et al. 2013; Nordemar et al. 2003), although other contemporary research has identified 557 balanced N:P enrichments as a controlling factor influencing coral response (Rosset et al. 2017; 558 Wiedenmann et al. 2013).

There is no contradiction between these studies. Replace “although other contemporary research has” by “In this context, the N:P ratio has been identified as controlling factor….”


“559 studies were characterized by exceptionally high DIN and/or DIP concentrations, which were 560 typically unbalanced and are unlikely to be found in natural systems.”

This is not true. See D’Angelo & Wiedenmann 2014 for scenarios that can lead to imbalance. Cite and discuss recent work by Lapointe 2019 https://link.springer.com/article/10.1007/s00227-019-3538-9 and 2020: https://www.sciencedirect.com/science/article/abs/pii/S0025326X21007207

Experimental design

Experimental design is ok.

Validity of the findings

See comment above regarding "Calcification rates"

Additional comments

n.a.

Reviewer 3 ·

Basic reporting

Han et al. conducted a manipulative experiment to investigate the combined effects of "minor" nutrient enrichment (nitrogen and phosphorus) and increased sea surface temperature on the bleaching susceptibility and calcification rates of three species of reef-building corals. The authors underlined delays in bleaching and greater calcification rates in corals exposed to both elevated nutrients and temperature compared to colonies under heat stress only - suggesting that nutrient enrichment may mitigate the deleterious effects of thermal stress on corals. In addition, Lobactis scutaria demonstrated a greater "remediation" from enrichment under elevated temperature compared to the other species.

Getting a better understanding of the interactive effects of nutrient enrichment and thermal stress on the physiological responses of reef-building corals is of prime importance - given the current status of coral reefs worldwide. The study by Han et al. thus contributes to increase our understanding on this timely topic. Overall, the manuscript is well written, structured and provides sufficient details regarding the experimental design and statistical analyses. It would have been great if the authors included additional measurements such as chlorophyll concentration or symbiont density in their study or photosynthetic rates. I have some comments regarding the literature references, the overall context of the experiment as well as the interpretation of the results and conclusions – which should be toned down given the limited measurements supporting the hypotheses. Please find my comments below.

Introduction:

The authors describe the current literature on the effects of nutrient enrichment on the physiological response of reef-building corals. They underlined the importance of a balance N:P ratio to maintain coral metabolism under nutrient and/or thermal stress and also describe the diverse effects of increased nitrogen vs phosphorus on coral physiology. Yet, the authors missed discussing the antagonistic effects of ammonium vs nitrate on coral physiology (Shantz & Burkepile, 2014 and Ezzat et al. 2015, 2019). This should be pointed out in the intro to complement the context of the study along with studies by Wiedenmann et al. 2013 and Rosset et al. 2017.

L.22: “algal recover” this term is unclear, please rephrase.
L.54: include reference
L.57: When specifying the different nutrient types, the authors need to discuss coral responses to elevated nitrogen vs phosphorus as well as the various effects of NH4 vs NO3 – See papers by Shantz & Burkepile, 2014; Ezzat et al. 2015 and Marangoni et al. xx.
L.64: Here you can also include Ezzat et al. 2019 – which investigated the effects of elevated NH4 concentrations and thermal stress on coral physiology.
L.72-79: DIN here represents NO3, please revise this section as well as the following 92-102 according to previous comments.
L.119-120: The authors will have to discuss the results by Wiedenmann et al. 2013. Wiedenmann et al. 2013 demonstrated that a balanced N:P including nitrate and phosphorus helped corals maintain their metabolism under thermal and light stress. The authors should tone down their statement here, the present study is not the first investigating the combined effects of low-lvl of nutrients and elevated sea surface temperature on coral physiology (see Becker et al. 2021 – mentioned earlier by the authors). The authors could elaborate a bit more about the originality of their studies based on the current literature.
L.131: The authors did not measure symbiont density (or I did not find the results?), please rephrase accordingly.

Experimental design

Material and methods

L.137: Please include the average size of coral colonies, not only their weight.
The authors could briefly describe the nutrient conditions in the reefs where the corals were found. What were the local nutrient concentrations and why did the authors decide to conduct a minor nutrient enrichment?

Looking at Table 1, it is unclear if the authors tested for differences in nutrient concentrations between all tanks. Did the concentrations in P differed significantly between A,H and N and NH treatments during the Heating phase? Same question for nitrogen.

Figure

Can the authors include a graph referring to the different phases of their experiment design, include the diverse time points at which measurements were taken? This would greatly help the readers.
Figure 5: Please include letters to indicate statistical significance

Validity of the findings

Discussion

L.442-449: The authors should tone down this statement.
L.467-469: What do the authors mean by “ameliorated the bleaching”? Its unclear.
L.485: Hard to compare studies like this without proper physiological measurements.
L.493-497: I do not fully agree with the authors here. Yes, phosphorus may have had a beneficial effect on coral physiology, potentially on the photosynthetic machinery and phospholipid membrane production. But the authors also exposed corals to increased NO3 and NH4. One could imagine that NH4 may have alleviated the potential deleterious effects of NO3 on coral physiology. This should be further discussed by the authors.
L.518: The fact that the authors noted differing response patterns according to coral species is interesting. I am wondering if the authors could elaborate a bit more on that matter especially from line L.552. What do the authors mean by “variable”? Are there any literature references describing the trophic plasticity of the different coral species used in this study -which could potentially explain the results obtained by the authors? Also please refer to the work of Houlebreque & Ferrier-Pagès, 2009 when discussing heterotrophy and corals.
In addition, the authors could elaborate a bit more on how different Symbiodinium clades could influence coral responses to nutrient and thermal stress.

Conclusion

Same comment as for the intro and discussion, the authors need to include the proper literature references to discuss their results.

---

## Round 0.2 · Minor Revisions

I have received comments back from one reviewer. Both the reviewer and I (see below) have some minor changes that should be incorporated or considered in a revised version of the manuscript.

L30 appearance should read coloration
L39 repossession of should read repopulation by
L49 have been largely should read have largely
L76 photobiology should read photosynthesis
L78 photobiology for corals should read coral photobiology
L172 diversion?
L525 relationships should read relationship
L531 partial conditions? Do you mean partial mortality?
L548 less reduced should read lower
L653, 655, 657, 658, 659 Symbiodinium should read Symbioidiniaceae

Reviewer 3 ·

Basic reporting

Some minor comments for the authors regarding the revisions they have made:

Line 112: These studies, however, only applied nitrate as a DIN enrichment source, which is typically not as preferential as ammonium and generally causes deleterious effects at higher concentrations. Indeed, antagonistic responses have occurred with differential nutrient identities (Ezzat et al. 2019; Marangoni et al. 2020).

It should read : Indeed, antagonistic responses have occurred with differential NITROGEN or DIN identities (Ezzat et al. 2019; Marangoni et al. 2020). Moreover, Ezzat et al. 2019 did not investigate the antagonistic effect of NO3 vs NH4, Ezzat et al. 2015 did.

Line 158: While several studies treated nitrate (NO3-) as the main DIN source, recent studies have found that the type of nutrient enrichment can result in different physiological responses. Contrasting responses of bleaching intensity under thermal stress have indeed been found, with positive effects under ammonium (NH4+) enrichment versus negative effects of nitrate (NO3-) enrichments (Ezzat et al. 2019; Marangoni et al. 2020).

What do the authors mean by the type of nutrients? It is the type of nitrogen... or the nitrogen identity (DIN) again here. Please replace accordingly.

Line 219: "The high nutrient, anoxic, and low light conditions prior to diversion led to an extreme decrease in coral cover and a simultaneous increase in filter or deposit feeders, phytoplankton, and algae (Hunter & Evans 1995; Laws & Redalje 1982). "

What does diversion mean in this context? Please rephrase

Line 869-875: "We assessed colony growth and various coral conditions (survivorship/mortality, partial conditions, bleaching progress) via detailed analyses of visual assessments under the low-level enrichment for 31 days of thermal stress. Overall, colonies under tested elevation (Nitrate: +0.51 μM, Ammonium: +0.75 μM, and Phosphate: +0.56 μM) showed no notable consequences with delayed signs of dysbiosis when compared to low nutrients under identical thermal stress. For all species tested, the onset of bleaching was delayed by 3.4 days in nutrient-enriched heated groups, which was likely due to the reduction in paled colonies midway through the heating period.

low-level enrichment ? something is off here. low-level nutrient
Overall, colonies under tested elevation - again something is off, please correct accordingly. What does "no notable consequences" mean?? physiological consequences?
The term "dysbiosis" appears here for the first time, please provide a clear definition of it.

Line 879: "This study is the only published account of prolonged exposure (>30 days) of coral to elevated temperature and minimal balanced nutrient enrichment. Other research involved short-term exposures to elevated temperature and (or) nutrients (Baker et al. 2018; Ezzat et al. 2016; Hadjioannou et al. 2019; Tanaka et al. 2014; Wiedenmann et al. 2013)."

What do the authors would like to underline, aside from the fact the study was conducted over a longer period compared to others?

Line 1061: "In this context, the N:P ratio has been identified as a controlling factor influencing coral response (Rosset et al. 2017; Wiedenmann et al. 2013)."

I agree with the authors, yet the nitrogen identity remains together with N:P an important factor that shapes coral health and physiological response under normal vs thermal stress conditions. This should be acknowledged as well.

Experimental design

-

Validity of the findings

-

---

## Round 0.3 · Minor Revisions

Thank you for making the suggested changes. I just have a few more suggested changes. Line numbers refer to the tracked changes verison.
1. L70 photobiology should read photosynthesis
2. L95 what do you mean by differential DIN identities (perhaps name them in parentheses)
3. L120 same as above but now called differential nitrogen identities.
4. L618-625 Please change all instances of Symbiodinaceae to Symbiodiniaceae

---

## Round 0.4 · accepted · Accept

I am satisfied with the changes made to the manuscript. The change to Symbiodiniaceae needs to be accepted in the second last paragraph of the discussion. The first changes was accepted but not the final three.